# Investigating the Effectiveness of Self-Regulated, Pomodoro, and Flowtime Break-Taking Techniques Among Students

**DOI:** 10.3390/bs15070861

**Published:** 2025-06-25

**Authors:** Eva J. C. Smits, Niklas Wenzel, Anique de Bruin

**Affiliations:** 1Faculty of Psychology and Neuroscience, Maastricht University, 6229 ER Maastricht, The Netherlands; 2Department of Educational Research & Development, School of Health Professions Education, Faculty of Health, Medicine, and Life Sciences, Maastricht University, 6211 LK Maastricht, The Netherlands; niklas.wenzel@maastrichtuniversity.nl (N.W.); anique.debruin@maastrichtuniversity.nl (A.d.B.)

**Keywords:** self-regulated break taking, pomodoro, flowtime, study experience, self-regulated learning

## Abstract

Effective break taking during study sessions is crucial for maintaining performance, especially in self-regulated learning settings where students plan their own tasks. However, research on break taking in these contexts is limited. This study investigates the effect of self-regulated, Pomodoro, and Flowtime breaks on subjective study experiences, task completion, and flow. Ninety-four university students participated in an online intervention that provided instruction on how to take breaks in a 2 h authentic study session. In the self-regulated break condition (*n* = 25), students decided when and how long to take breaks. In the Pomodoro condition (*n* = 36), students took 5 min breaks after every 25 min of studying. In the Flowtime group (*n* = 33), participants decided when to take breaks; however, the break duration was determined based on prior study duration. Results showed that Pomodoro breaks led to a faster increase in fatigue, and Pomodoro and Flowtime breaks led to a faster decrease in motivation compared with self-regulated breaks; however, these differences did not result in overall differences in fatigue or motivation levels between conditions. Similarly, no differences were found in productivity levels, task completion, and flow. Future research should further examine these techniques by including variables like personality and mental effort.

## 1. Introduction

Effective learning is a demanding and effortful process that can lead to feelings of stress and decreases in productivity levels and well-being ([15]; [16]). Recovery from the demands of study tasks is essential for maintaining high levels of performance during the day ([31]). Taking breaks from study tasks allows for the opportunity to recover personal resources (e.g., motivation and energy levels), which are necessary for an effective study session ([42]). Taking breaks may therefore counteract the negative effects of prolonged task engagement and may help to complete tasks more efficiently ([26]; [28]; [31]). The specific timing and duration of breaks may have differential effects on the recovery of these resources ([27]; [42]). Moreover, taking effective breaks can enhance the study experience of students and may improve their well-being and mental health ([25]; [31]). Despite the benefits of effective break taking, some challenges students face in self-regulating these breaks are still not fully understood.

One of these challenges is how students can effectively regulate their breaks, especially in contexts outside the classroom, where students must manage their learning independently. Self-regulated learning (SRL) is defined by [32] ([32]) as “an active, constructive process whereby learners set goals for their learning and then attempt to monitor, regulate, and control their cognition, motivation, and behavior, guided and constrained by their goals and the contextual features in the environment” (p. 453). SRL is thus an independent and self-directed process in which students are fully responsible for regulating their learning ([17]). In these settings, students are therefore required to make decisions about when, where, and how to study. A lot of students struggle with self-regulating their learning because of difficulties in attention retention, motivation, and lack of metacognitive knowledge ([3]; [36]; [45]; [46]).

Deciding to engage in study tasks involves the decision to forgo other potentially rewarding activities, such as socializing or going outside. According to the opportunity costs model of [22] ([22]), this trade-off constitutes opportunity costs. The model suggests that the perceived costs of a task are influenced by the appeal of other available opportunities and the perceived benefits of investing effort in the task at hand ([22]). For example, studying for an exam may be perceived as less effortful if a student expects it will lead to a higher grade. Taking a break from study tasks provides an opportunity to engage in alternative activities, potentially reducing the perceived effort and restoring motivation ([12]). In this way, break taking may support learning by restoring personal resources. Break taking can, but is not necessarily, part of self-regulated learning, as students’ break taking may impact the achievement of their learning goals ([28]). There are different techniques for regulating breaks during self-regulated learning. Some students may plan breaks in advance, using techniques that externally regulate breaks, while others rely on subjective cues such as fatigue, productivity, and perceived effort and progress productivity to decide when to disengage from learning tasks ([5]; [9]; [18]).

Given the challenges students face while self-regulating their learning, it is important to understand how students can improve their study technique by effectively regulating their break taking to enhance their study experience. Therefore, additional research on the effectiveness of different break-taking techniques in self-regulated learning settings is needed. The following section outlines different break regulation techniques and their potential implications for student learning.

## 2. Types of Break Regulation Techniques

### 2.1. Self-Regulated Breaks

In self-regulated break taking, students take breaks whenever they want. The decision to take a break is based on a subjective feeling and is therefore not planned. The term “self-regulated breaks”, as used in this paper, refers to taking breaks whenever a student feels like it. Students also determine for themselves when and how long to take a break. A commonly described disadvantage of self-regulated breaks is that they can increase the cognitive load by inducing an additional self-monitoring load. The mental effort that is used for this cannot be used for studying, which can make studying more cognitively demanding ([24]).

In addition, deciding yourself when to take a break is a form of self-interruption. Self-interruptions are interruptions that are initiated internally ([20]). Contrarily, external interruptions are interruptions that occur due to something outside of the self ([1]; [20]). In self-regulated breaks, individuals interrupt their study tasks in response to an internally driven desire for a break. Research has shown that self-interrupted tasks are completed more slowly than externally interrupted tasks ([20]). Due to these self-interruptions, self-regulated break taking may reduce motivation and productivity, which may prolong the time needed to complete study tasks ([20]).

To minimize the additional mental load and self-interruptions, various break-taking techniques have been developed that are thought to enhance effectiveness; however, research on these techniques in self-regulated learning settings is limited.

### 2.2. The Pomodoro Technique

The Pomodoro technique is a well-known technique, as it is frequently promoted through online platforms (e.g., [37]; [40]). When using the Pomodoro technique, students study in blocks of 25 min with a 5-min break in between. These blocks are repeated four times before taking a longer break of fifteen to thirty minutes ([38]). In this case, breaks are externally regulated by a timer ([11]). Even though the break duration and timing are regulated externally, students still have to self-regulate their own learning and regulate their breaks by setting the timer. Systematic breaks like Pomodoro are thought to postpone distractions and facilitate motivation and concentration. In addition, Pomodoro breaks are thought to improve productivity and reduce mental fatigue. The Pomodoro technique might specifically diminish procrastination because it requires students to split big tasks into smaller sections, which can lead to a higher degree of task completion ([2]). The Pomodoro technique is therefore often recommended online, even though it has received surprisingly little attention in education research.

[4] ([4]) conducted a study about systematic break taking and its effect on mental effort, task completion, and study experiences. In this study, three independent conditions were included, one self-regulated break-taking group and two systematic break-taking groups. One condition was similar to Pomodoro, and the other systematic break-taking group used even smaller study blocks of 12 mininute with a 3 mininute break in between. They found that students in both systematic break conditions reported being more concentrated and motivated, and perceived learning tasks to be less difficult, than in the self-regulated group ([4]).

Even though the Pomodoro technique is thought to have a lot of advantages, one critique is that this technique can disrupt the so-called flow state. This is a state in which someone is highly focused and productive. In this state, people usually do not feel time passing and feel absorbed in their tasks ([8]). For a student to experience flow, complete involvement in the current task as well as the loss of awareness of everything outside of the task is necessary ([43]). The flow state is very fragile; for example, this state can already be disrupted when a small distraction is noticed ([43]). A flow state can therefore already be interrupted by a thought about a grade or a small mistake ([44]). Because Pomodoro uses a timer during study time and is very strict about when to take a break, this technique can disrupt the flow state ([41]).

### 2.3. The Flowtime Technique

A break-taking technique that might overcome this problem is the Flowtime technique. The technique is also known as the “Flowmodoro” technique, since it is a variation of the Pomodoro technique ([39]). In the Flowtime technique, students study for as long as they can focus and take a break whenever they feel the need. The duration of the break depends on how long they have worked on their study tasks. For example, if someone studies for 25 min or less, they will receive a 5 min break, like in the Pomodoro technique. However, if someone decides to study for a longer period of time, they will receive a longer break ([23]). This technique combines systematic elements of the Pomodoro method and self-regulatory elements of the self-regulated breaks. This combination may, on the one hand, lead to a disadvantage compared with the Pomodoro technique since students still need to self-regulate when to take a break, which may increase cognitive load, which can make studying more demanding ([24]). However, since students do not have to determine the length of their breaks themselves, cognitive demands are still not as high as with completely self-regulated breaks. On the other hand, the self-regulated element in the Flowtime technique leads to a considerable advantage over the Pomodoro technique. Since the Flowtime technique does not use a timer during study time, the flow state will not be interrupted, and it therefore resolves the critique of the Pomodoro technique ([30]).

## 3. The Present Study

This study aims to examine the differences between the self-regulated, Pomodoro, and Flowtime break techniques and their effectiveness during self-regulated learning in higher educational settings. The relation between break-taking technique and motivation, fatigue, productivity, task completion, and flow will be investigated during an authentic self-regulated learning session. Therefore, the present study is centered around the following question: “What is the effect of the type of break-taking technique (self-regulated, Pomodoro, or Flowtime breaks) on motivation, productivity, fatigue, task completion, and flow state?” To test this research question, several hypotheses were formulated:

**Hypothesis** **1.**
(a)
*The Pomodoro and Flowtime conditions will show higher levels of motivation compared with the self-regulated break condition.*
(b)
*The Pomodoro and Flowtime conditions will show higher levels of productivity compared with the self-regulated break condition.*
(c)
*The Pomodoro and Flowtime conditions will show lower levels of fatigue compared with the self-regulated break condition.*



**Hypothesis** **2.**
*The Pomodoro and Flowtime conditions will show higher levels of task completion compared with the self-regulated break condition.*


**Hypothesis** **3.**
*The flow state will be greater in the self-regulated break and Flowtime conditions than in the Pomodoro condition.*


Furthermore, this paper examines whether different break-taking techniques influenced changes in subjective study experiences over time and whether individuals exhibited distinct trajectories over time. This is addressed by a second exploratory research question: “How do different break-taking techniques (self-regulated, Pomodoro, and Flowtime breaks) influence the trajectory of students’ subjective study experiences (namely motivation, productivity, and fatigue) during a study session, and are there differences in these changes between individuals?”

## 4. Materials and Methods

### 4.1. Participants and Design

The sample of this study included 111 psychology students from a Dutch university, which was deemed necessary based on the less advanced analysis method initially planned, as the study was conducted for a bachelor’s thesis. Since a more advanced, better-suited analysis method is used in the current paper, the required sample size for achieving sufficient power might have been higher. Ethical approval was obtained from the ethical board of the university (The Ethical Review Committee of Psychology and Neuroscience, ERCPN) with the corresponding reference code ERCPN-274_120_11_2023. Participants were recruited online through the SONA system, which is a system that allows universities to create participant pools and upload studies.

Participants were randomly assigned to one of three conditions: self-regulated, Pomodoro, or Flowtime breaks. All groups received exactly two hours to study and take breaks. The time students studied between one break and the next break is called a “study block”. Students in the self-regulated group took self-regulated breaks and were allowed to study and take breaks as they felt like doing. Participants in this group were instructed to study as long as they wanted and afterward take a break as long as they needed. The second group used the Pomodoro technique for taking breaks and was instructed to study in four blocks of 25 min with 5 min breaks in between. To ensure consistency across all groups, participants in this condition did not receive the extended break after the fourth study block. Providing these students with a longer final break would have exceeded the two-hour limit. Moreover, since participants in the other conditions did not consistently end their sessions with a break, this could have biased the results of this study. The last group used the Flowtime technique and was therefore instructed to study as long as they could focus, and they were allowed to take a break if they could not focus any longer, as in the self-regulated group. The duration of the break these participants received was dependent on the duration of the previous study block. Students received a 5-min break after studying for 25 min or less, and an 8-min break if they studied between 25 and 50 min. If students studied for more than 50 min after the last break, they received a 10-min break. The precise instructions that participants received per condition can be examined in Appendix A.

### 4.2. Measures

Data collection and survey distribution occurred via the Qualtrics survey software. Participants could access the Qualtrics survey via SONA. The full questionnaire with all measures described below is provided in Appendix B.

#### 4.2.1. Subjective Study Experience

Motivation, productivity, and fatigue were measured in a similar way as in the study by [4] ([4]). This was accomplished with questions like, “How motivated do you feel right now?”. In the study by [4] ([4]), these questions were answered on a 5-point Likert scale. However, due to a human error, the questions were answered by the participants on a 4-point Likert scale in this study. Every participant answered one item per measure (i.e., motivation, productivity, and fatigue) before starting the study session, after every break, and after the two hours of study.

#### 4.2.2. Task Completion

Task completion was also measured in a similar way as in the study by [4] ([4]). Participants indicated at the beginning of the study session what tasks they were planning to work on. Participants had to indicate at least one task and could indicate up to eight tasks. After the 2 h study session, the participants indicated for each task to what extent they had completed the task on a scale from 0 to 100 percent. To compute the total task completion, the mean of these completion percentages across all tasks was calculated.

#### 4.2.3. Passed Time and Break Duration

Time was measured using the timing option in Qualtrics. This option measures the time a student is at a particular part of the questionnaire, i.e., at the “Time for study” or at the “Time for a break” part. The participants were instructed to keep the Qualtrics survey open during the whole study period. The participants could thus use their laptops for their study tasks but could leave the questionnaire open in the background. By doing this, the Qualtrics software automatically measured the time students spent studying and taking breaks.

#### 4.2.4. Flow State

Lastly, a flow questionnaire was used to measure to what extent participants experienced they were in a flow state during the 2 study hours. For this, the Flow short scale (FSS) was used, which is a 10-item questionnaire designed to assess the state of flow. It utilizes a 7-point Likert scale ranging from 1 (not at all) to 7 (very much). This scale was designed in German and translated by [33] ([33]) in English. This version of the questionnaire is reliable and is supported by evidence of validity for use in flow state research with a Cronbach’s alpha of 0.90 ([33]). The English version was used for this study, and participants filled it in after the study session. The total flow state was calculated by adding up the scores on all items.

### 4.3. Procedure

Data collection for the current study took place between January and March 2024. Figure 1 provides an overview of the procedure of the study. Before signing up, participants were informed that the research focused on break taking among students and that it would take approximately 120 min to complete. Participants were also informed that the study would be carried out online. After signing up, they were given the opportunity to read the information letter with information about the research and procedure and gave informed consent before the start of the study.

The study relied on an experimental design commonly used in cognitive psychology, but it was conducted in an authentic study setting. Students could engage with their own academic materials with minimal interference from the research. Participants first completed a short questionnaire about their current motivation, productivity, and fatigue levels, planned tasks, and some demographic questions. After this, all participants were randomly assigned to one of three conditions. In all three groups, participants were instructed to work on the study tasks that they had intended to work on in the following study block of two hours. Subsequently, the three groups received distinct instructions about taking breaks; the precise instructions that the participants received are provided in Appendix A.

The self-regulated group was instructed to press a button in the Qualtrics environment when they started and ended their break. For the Pomodoro group, the questionnaire automatically switched between study time and break time. An audio signal informed participants when to take a break and when to continue studying. The Flowtime group was asked to press a button when starting their break, and a sound was played to inform them that they should start studying again.

In addition to hearing a sound, every participant was instructed on the screen what they should be doing at a specific time. The screen all participants saw while studying stated, “Start to study!” and the break screen stated, “Time for a break!” In the Pomodoro and Flowtime groups, the break screen also indicated the total duration of the current break.

After every break, participants in all conditions were asked to indicate how motivated, fatigued, and productive they felt. Furthermore, immediately after the two hours of self-regulated learning were completed, the participants were directed to a last set of questions about their subjective study experiences, task completion, and flow state. For some participants the questionnaire failed to transfer to the last part of the questionnaire automatically; these participants were asked to click a link that would directly lead to the last set of questions after the two hours had passed.

### 4.4. Analysis

We tested three main sets of hypotheses regarding the effects of different break-taking techniques on motivation, productivity, fatigue, task completion, and flow. Outcome measures were collected at different time points. Motivation, productivity, and fatigue were assessed repeatedly on a 4-point Likert scale. For these outcome variables, we used cumulative link mixed effects models (CLMMs) to account for the ordinal nature of the data as well as the unbalanced repeated measures design. In contrast, task completion (calculated as a percentage of completed tasks) and flow (assessed using 10 items on a 7-point Likert scale) were measured only once at the end of the study period. For these latter outcomes, we aggregated the repeated measures (using means for the motivational and productivity variables) and applied linear regression analysis.

#### 4.4.1. Preparing the Analysis

Data cleaning and preparation were performed prior to the main analyses to ensure data quality, adherence to study parameters, and suitability for the planned statistical tests. Participants were excluded if their data did not meet the predetermined criteria necessary for valid analysis. Specifically, exclusions were made based on non-adherence to study time parameters. This involved removing individuals whose total recorded time exceeded a cutoff of 130 min (*n* = 9), thereby excluding prolonged participation potentially indicative of off-task behavior. Additionally, this cutoff allowed us to account for natural inefficiencies that characterize authentic learning environments, such as delayed switches from break to studying and vice versa. Participants were also excluded if their engagement with their self-selected study tasks was minimal (study time < 10 min, *n* = 2) or if their break time was excessive (total break time > 110 min, *n* = 1), as this indicated insufficient engagement to provide meaningful data relevant to our research questions. Furthermore, exclusions were necessary when participants’ data were unsuitable for the core analyses central to our study. This included participants with incomplete repeated measures data (*n* = 2), which prevented their inclusion in longitudinal analyses, and those who took no breaks during the study session (*n* = 2). As our analyses focused specifically on the characteristics and effects of breaks, data from this latter group could not contribute to testing these specific hypotheses. Finally, one participant was identified as a statistical outlier whose overall data pattern suggested they were not representative of the target sample or could disproportionately influence results. This participant was initially flagged as an outlier via IQR on age (age = 38) and was excluded because they also showed divergent patterns on multiple key study engagement variables, indicating an anomalous response profile. In total, 17 participants were removed based on these combined criteria related to study adherence, data completeness, analytic requirements, and anomalous patterns, reducing our analysis sample from 111 to 94 participants.

The final sample’s characteristics were n = 94, with a mean age of 20.02 years (*SD* = 1.87) and 87 (93%) women. When broken down by group, the self-regulated group included 25 participants (mean age 19.84, *SD* = 1.60; women = 22), the Flowtime group consisted of 33 participants (mean age 20.20, *SD* = 1.98; women = 31), and the Pomodoro group had 36 participants (mean age 20.00, *SD* = 1.97; women = 34). All variables were recoded and designated as the appropriate type within R (continuous, factor, etc.) before model estimation.

#### 4.4.2. Analysis Methods

To test our hypotheses, we employed two main modeling strategies that corresponded to the distinct measurement structures in our study. As mentioned above, for the repeatedly measured outcomes (motivation, productivity, and fatigue, each assessed on a 4-point Likert scale), we used CLMMs. This choice was driven by the ordinal nature of the data and the need to account for unbalanced repeated measures within participants, which CLMMs can account for. Starting with fully saturated models that incorporated theoretically relevant predictors and random effects, we iteratively simplified these models based on likelihood ratio tests, information criteria (AIC), and theoretical considerations until arriving at the final, robust specifications. After the final model specification, the hypothesis was tested by testing nested models (with and without the predictor in question) using likelihood ratio tests and information criteria (AIC). All relevant assumptions were checked (see below).

For outcomes that were measured only once as post-measures (task completion and flow), we aggregated repeated measures (using means) where needed and applied linear regression. In these analyses, standard assumptions (normality of residuals, linearity, homoscedasticity, independence, and multicollinearity) were evaluated using both visual and statistical measures (e.g., Q-Q plots, Shapiro–Wilk, Breusch–Pagan, and Durbin–Watson tests with a *p* > 0.05 threshold) to ensure model adequacy. Finally, we tested for potential influential cases by visually inspecting the residual Q-Q plots for each model and by using the interquartile range criteria for continuous variables. We then re-fitted the final models for each hypothesis with these potential influential cases removed. In most cases, this did not lead to a significant difference. The exception to this was the test for H1c, where the removal of two cases at the extreme negative end of the Q-Q plot led to the number of breaks predictor becoming significant.

This general modeling approach—CLMMs for ordinal repeated measures and linear regression for aggregated single-point outcomes—provided a structured and theoretically informed foundation for testing our hypotheses. Specific details regarding model specifications and assumption diagnostics are provided in the respective results sections.

#### 4.4.3. Model Building and Assumption Checks

For the cumulative link mixed model (CLMM) analyses, we followed a structured model-selection process, beginning with the most complex models deemed theoretically plausible for each hypothesis and aiming to identify the best-fitting, most parsimonious model. We started by determining the most appropriate linking function for the ordinal outcome variable for each hypothesis by comparing probable link functions using AIC and BIC. We compared the logit and the loglog function. The former is often the default choice when it comes to fitting ordinal models and models with a symmetric relationship between predictors and the response variable. We chose to compare this logit function against the loglog function because some of our outcome variables showed a slight tendency toward a left skew, and the loglog function can account for this. Once we established the appropriate linking function, we determined the optimal random effects structure. Our initial complex structure included a random intercept for participant and random slopes for the duration of the last break. We chose this because the duration of the break was one of the variables that could vary between groups, and adding it as a random slope allowed us to assess whether individuals responded differently to the duration of the break per se (beyond the effect of group). We systematically evaluated this more complex model against simpler, nested alternatives (e.g., random intercept only) using likelihood ratio tests (LRTs) for nested models supplemented by AIC and BIC for converging evidence. When criteria conflicted, we favored the more parsimonious model. With the linking function and random effects structure established, we simplified the fixed effects using a backward elimination approach. Starting with the full model containing all theoretically relevant predictors and their interactions, we sequentially tested the removal of terms, beginning with higher-order interactions, using LRTs and changes in AIC/BIC. Non-significant terms were removed unless they were main effects involved in significant or marginally significant interactions. Throughout this simplification, the centered time-passed variable and group variable were always retained to account for the repeated-measures design and due to their centrality to our hypotheses. This iterative process continued until only statistically significant predictors, terms involved in significant interactions, or theoretically essential predictors remained, resulting in the final model used for hypothesis testing.

We used several methods to test the relevant assumption for the CLMMs. A visual inspection of Q-Q plots was used to verify that the random effects approximated a normal distribution. The proportional odds assumption was evaluated via nominal test (using clm() and the nominal_test() function of the ordinal package (v2023.12-4.1; [7]) with *p* > 0.05 taken as evidence of no violation. This meant that we had to refit the final model without the random effects structure because the clmm() function was not supported by the nominal_test() function at the time of writing. Importantly, H1b showed a potential violation of the proportional hazards assumption. Visual checks using cumulative probability plots (cumulative probabilities for each level of a given predictor plotted against the outcome thresholds) indicated minor departures for the group and fatigue predictors at the lowest threshold of the outcome variable (productivity). Additionally, the lines of cumulative probabilities across the levels of each predictor tended to converge at the highest level of the outcome variable. Based on these evaluations, we assumed that the proportional hazards assumption is mostly intact but that the findings related to productivity as outcome should be treated with care. Finally, models without the random effects structure were fitted and compared (via likelihood ratio tests and AIC) to ensure that the inclusion of random effects improved the model fit.

For the linear regression models, we conducted standard assumption checks. The normality of individual variables and of residuals was confirmed via visual Q-Q plots and the Shapiro–Wilk test (with *p* > 0.05 indicating acceptable normality). Linearity was checked through residual plots. Homoscedasticity was assessed with both a visual inspection of the residuals and the Breusch–Pagan test (again using *p* > 0.05 as supportive evidence). Independence of residuals was evaluated using the Durbin–Watson test. Multicollinearity was examined by calculating variance inflation factors (VIFs) using the car package. None of these tests indicated any violation of assumptions for H2 (task completion) or H3 (flow). In addition, potential influential cases were identified via visual inspection of residuals and re-fitting the models. In neither case did this change the results significantly.

## 5. Results

The means and standard deviations of variables measured in the current study can be found in Table 1.

### 5.1. Motivation

Hypothesis 1a posited that the Pomodoro and Flowtime groups would report higher levels of motivation relative to the self-regulated group. To test this, we fitted a CLMM model predicting motivation scores (assessed on a 4-point Likert scale). The final model included fixed effects for group membership; centered passed time since the start of the study; linear and quadratic orthogonal polynomial terms for lagged fatigue, lagged productivity, and lagged motivation; and the interaction between group membership and centered passed time. Participant-level random intercepts were included to account for the non-independence of repeated measures within individuals. Table 2 summarizes the model estimates.

Model evaluation indicated that while the random intercept variance (0.54, *SD* = 0.73) suggests some individual differences in baseline motivation log-odds, a model without random effects (CLM) provided a slightly better fit according to BIC (846.0 vs. 849.4) and was not significantly worse according to a likelihood ratio test (*χ*^2^(1) = 2.46, *p* = 0.117), indicating that the effect of the duration of the last break before the repeated measurement did not differ significantly between participants. However, the CLMM was retained to explicitly account for the nested data structure inherent in the repeated measures design. Random effects appeared normally distributed based on a Q-Q plot. The final model showed a significantly better fit than a null model including only random intercepts and thresholds (*χ*^2^(11) = 89.78, *p* < 0.001, BIC = 849.4 vs. 874.4). Examination of the proportional odds assumption using a nominal test on a fixed-effects equivalent model and visual inspection of coefficients across binary splits suggested no major violations, although the linear term for lagged motivation showed some deviation at the lowest threshold, warranting caution in its interpretation.

Threshold coefficients defining the latent motivation levels were estimated as follows: 1|2: −2.87 (*SE* = 0.36), 2|3: −0.59 (*SE* = 0.27), 3|4: 2.35 (*SE* = 0.31).

The participant-level random intercept variance was 0.54 (*SD* = 0.73), indicating notable variability in baseline motivation between participants.

In terms of hypothesis testing, the group comparisons did not reach statistical significance at the reference point of time passed (the midpoint of the study session). Neither the Pomodoro group (*OR* = 1.55, β = 0.44, *SE* = 0.34, *p* = 0.200) nor the Flowtime group (*OR* = 0.98, β = −0.02, *SE* = 0.35, *p* = 0.953) differed significantly in motivation levels from the self-regulated group. However, significant interactions were found between group membership and the centered time variable. To explore this, we calculated the estimated marginal means and compared the slopes of the three groups. This analysis showed that, compared with the self-regulated group, both the Pomodoro group (*OR* = 0.98, β = −0.02, *SE* = 0.01, *p* = 0.008) and the Flowtime group (*OR* = 0.99, β = −0.01, *SE* = 0.01, *p* = 0.039) showed a significantly steeper decline in the odds of higher motivation as the study session progressed, indicating that the odds of being in a higher level of motivation reduced by ca. 2% and 1% more per minute for the Pomodoro and Flowtime group, respectively. Post hoc analyses confirmed these interaction effects. The analysis of slopes (see Figure 2) indicated that the rate of change in motivation log-odds as time passed was significantly more negative for the Pomodoro group compared with the self-regulated group (Δ_β_ = 0.01, *SE* = 0.01, *p* = 0.024, Bonferroni-adjusted). The slope for the Flowtime group was also more negative than that of the self-regulated group (Δ_β_ = 0.02, *SE* = 0.01, *p* = 0.118, Bonferroni-adjusted), but it did not reach statistical significance after Bonferroni adjustment.

Regarding the lagged subjective experience variables, the model included orthogonal quadratic polynomial terms. Higher prior fatigue was associated with lower current motivation log-odds, with significant effects for both the linear (*OR* = 0.01, β = −4.97, *SE* = 2.73, *p* = 0.069) and quadratic terms (*OR* = 0.01, β = −4.43, *SE* = 2.36, *p* = 0.061), indicating a non-linear relationship with approaching significance at the *p* < 0.05 level. Lagged productivity showed a significant positive linear association (*OR* = 1060.77, β = 6.97, *SE* = 2.95, *p* = 0.018), indicating higher prior productivity predicted higher current motivation log-odds; the quadratic term was not significant (*p* = 0.604). Lagged motivation itself showed a strong positive linear association (*OR* = 55710.93, β = 10.93, *SE* = 3.75, *p* = 0.004), indicating motivational persistence from the previous time point; the quadratic term was not significant (*p* = 0.403).

Overall, while the hypothesis of consistently higher mean motivation in the Pomodoro and Flowtime groups compared with the self-regulated group was not supported by the main effects, the significant interaction effects reveal a crucial dynamic: participants using the Pomodoro technique and the Flowtime technique experienced a more pronounced decline in motivation as time elapsed since their last break compared with the self-regulated learners. Furthermore, the significant effects of prior fatigue, productivity, and motivation levels underscore the importance of these dynamic state variables in predicting subsequent motivation during study sessions. It is important to point out that the strong effects of the lagged predictors for productivity and motivation might indicate an issue of (partial) separation, potentially driven by the low number of observations at the extremes of the outcome variable. The effect of these predictors, and especially their magnitude, should therefore be treated with great care.

### 5.2. Productivity

Hypothesis 1b posited that the Pomodoro and Flowtime groups would report higher levels of productivity relative to the self-regulated group. To test this, we fitted a cumulative link mixed effects model with a logit link function. The final model included fixed effects for group membership; time (centered); orthogonal quadratic polynomial terms for lagged fatigue, lagged productivity, and lagged motivation; and the interaction between group membership and the time variable. Participant-level random intercepts were included to account for repeated measures. Table 3 summarizes the key fixed effects estimates from the model.

The model evaluation revealed that the random intercept variance (*σ*^2^*_intercpt_* = 0.16, *SD* = 0.40) was relatively small. A comparison with a model lacking random effects (CLM) showed the CLM was preferred by AIC (785.7 vs. 787.4) and BIC (840.2 vs. 845.7), and the likelihood ratio test was non-significant (*χ*^2^(1) = 0.36, *p* = 0.546), suggesting the random intercept did not significantly improve model fit, indicating that participants likely do not differ significantly in their level of productivity felt. Despite this, the CLMM was retained to explicitly model the dependency of repeated measures within individuals. Random effects appeared normally distributed via Q-Q plot inspection. The final model provided a significantly better fit than a null model containing only thresholds and the random intercept (*χ*^2^(11) = 72.52, *p* < 0.001). Tests of the proportional odds assumption (using a fixed-effects equivalent model) indicated no significant violations overall, although lagged fatigue (*p* = 0.064) and lagged motivation (*p* = 0.053) showed marginal significance, suggesting potential deviations, particularly at the lowest threshold where data might be sparser. Interpretation, especially concerning fatigue and motivation effects on transitions into the lowest productivity levels, should proceed with this caution.

Threshold coefficients that define the latent productivity categories were estimated as follows: 1|2: Estimate = −3.38 (*SE* = 0.35), 2|3: Estimate = −1.24 (*SE* = 0.25), and 3|4: Estimate = 1.41 (*SE* = 0.25). The participant-level random effects indicated a variance of 0.16 (*SD* = 0.40) for the intercept, indicating relatively low variability in baseline productivity log-odds between participants after accounting for fixed effects. This is in line with the better fit of the CLM model that was discussed above.

In terms of hypothesis testing, the main effects for group comparisons were not statistically significant, indicating that groups did not differ in perceived productivity in the middle of the study session (average of centered time). Neither the Pomodoro group (*OR* = 0.97, β = −0.04, *SE* = 0.29, *p* = 0.904) nor the Flowtime group (*OR* = 0.75, β = −0.28, *SE* = 0.30, *p* = 0.348) differed significantly in productivity log-odds from the self-regulated group at this reference point. The main effect for time was also not significant (*OR* = 1.00, *β* = 0.00, *SE* = 0.01, *p* = 0.739). The interaction between the Pomodoro group and time since the last break was not significant (*OR* = 0.99, β = −0.01, *SE* = 0.01, *p* = 0.150). A marginally significant interaction was observed, however, between the Flowtime group and centered time (*OR* = 0.99, β = −0.01, *SE* = 0.01, *p* = 0.051). This suggests that for participants in the Flowtime group, the log-odds of higher productivity tended to decrease more steeply over time compared with the self-regulated group. Estimated marginal means were calculated and post hoc analyses were performed. Specifically, pairwise comparisons of the group slopes for the log-odds of productivity over time showed that the slope for the Flowtime group was more negative than that for the self-regulated group, consistent with the interaction term. Yet, this difference did not reach statistical significance after Bonferroni adjustment (∆_β_ = 0.01, *SE* = 0.01, *z* = 1.96, *p* = 0.152). Figure 3 shows the slopes for the three groups across time.

Regarding the lagged state variables, the linear effect of lagged fatigue was significant (*OR* = 0.01, β = −5.26, *SE* = 2.45, *p* = 0.032), indicating that higher prior fatigue was associated with lower current productivity log-odds. The quadratic term for fatigue was not significant (*p* = 0.798). Both lagged productivity (Linear: *OR* = 116,662.27, β = 11.67, *SE* = 3.43, *p* < 0.001; Quadratic: *p* = 0.244) and lagged motivation (Linear: *OR* = 17,326.63, β = 9.76, *SE* = 2.88, *p* < 0.001; Quadratic: *p* = 0.602) showed strong, significant positive linear associations with current productivity log-odds. This suggests that higher productivity and higher motivation in the previous assessment strongly predicted increased productivity in the current assessment.

Overall, the hypothesis of direct group differences in productivity was not supported. There was a trend (*p* = 0.051) suggesting that productivity in the Flowtime group decreased more rapidly than in the self-regulated group as time since the last break increased. Furthermore, the results strongly underscore the importance of immediate history: lower prior fatigue, higher prior productivity, and higher prior motivation were all significant predictors of higher current productivity levels. As with the model for motivation, the findings in relation to the lagged productivity and motivation predictors should be handled with care due to potential separation issues, potentially driven by sparse data at the extremes of the outcome variable.

### 5.3. Fatigue

Hypothesis 1c predicted that participants in the self-regulated group would report higher fatigue compared with those in the Pomodoro and Flowtime groups. To test this, we modeled fatigue, assessed on a 4-point Likert scale, using a cumulative link mixed effects model with a loglog link function (please note that we are not representing odds ratios in this section, due to the nature of the loglog link function).

The final model included fixed effects for group membership, time (centered), duration of the last break (centered), orthogonal quadratic polynomial terms for lagged fatigue and lagged productivity, and the interaction between group membership and time. The model incorporated participant-level random intercepts and random slopes for the effect of centered break duration to account for within-subject variability and individual differences in the break duration effect. Table 4 summarizes the key fixed effects estimates from the model.

The model evaluation indicated that while the random effects structure was retained based on model-building steps, formal comparisons via AIC (714.2 vs. 709.6), BIC (776.4 vs. 760.1), and LRT (χ^2^(3) = 1.38, *p* = 0.709) suggested a simpler model without random effects (CLM) might offer comparable or slightly better fit based purely on these metrics. However, the CLMM was retained to explicitly account for the nested data structure inherent in the repeated measures design. The random intercept variance was relatively small (σ^2^_intercpt_ = 0.07, *SD* = 0.26), as was the random slope variance for break duration (σ^2^_slope_ = 0.00, *SD* = 0.06). A perfect negative correlation (−1.00) was estimated between the random intercept and slope, which might indicate boundary estimation issues, though the model converged successfully. Q-Q plots of random effects showed reasonable normality; although potential outliers were noted, they did not significantly alter results upon removal. The final model significantly outperformed a null model (χ^2^(12) = 103.48, *p* < 0.001). Tests of the proportional odds assumption (using a fixed-effects equivalent model) showed no significant violations overall, but the quadratic term for lagged fatigue could not be assessed directly in the nominal test output, and visual checks suggested potential non-proportionality at the lowest threshold, warranting caution in interpreting this specific term. The threshold coefficients that delineate the latent fatigue categories were estimated as 1|2: −1.55 (*SE* = 0.18), 2|3: −0.13 (*SE* = 0.15), and 3|4: 2.36 (*SE* = 0.22). The participant-level random effects estimates were intercept variance = 0.065 (*SD* = 0.26), break duration slope variance = 0.003 (*SD* = 0.06), correlation = −1.00.

Regarding Hypothesis 1c, the main effects of group were not statistically significant at the average level of centered time (the mid-point of the study session), indicating that baseline fatigue log-odds did not differ markedly between the self-regulated, Pomodoro (β = −0.11, *SE* = 0.19, *p* = 0.558), and Flowtime groups (β = −0.30, *SE* = 0.19, *p* = 0.117). The main effect for centered time since the last break was also not significant (β = 0.00, *SE* = 0.00, *p* = 0.885). However, a significant interaction emerged between the Pomodoro group and centered time since the last break (β = 0.01, *SE* = 0.00, *p* = 0.025). This indicates that, compared with the self-regulated group, the log-odds of reporting higher fatigue increased more steeply for participants in the Pomodoro group as more time elapsed since their last break. The interaction between the Flowtime group and time since the last break was not statistically significant (β = 0.00, *SE* = 0.00, *p* = 0.577).

Post hoc analyses confirmed the interaction pattern. Pairwise comparisons of the slopes showed that the rate of increase in fatigue log-odds over time was steeper for the Pomodoro group compared with the self-regulated group (∆ = 0.01, *SE* = 0.00, *z* = 2.25, *p* = 0.074), although this comparison did not reach significance after Bonferroni adjustment. No other slope comparisons were significant after adjustment. Pairwise comparisons of estimated marginal means at specific time points revealed that at the third quartile of time since the last break (ca. 120 min), the Pomodoro group had significantly higher estimated fatigue log-odds than the Flowtime group (∆ = 0.50, *SE* = 0.19, *z* = 2.60, *p* = 0.028, Bonferroni-adjusted). Figure 4 shows the slopes for the three groups across time. 

Regarding other predictors, the centered duration of the last break did not show a significant fixed effect on fatigue (β = −0.04, *SE* = 0.03, *p* = 0.172), although its effect was kept as a random slope based on improved model fit (as indicated by AIC and BIC). Lagged fatigue showed a strong positive association with current fatigue, significant for both the linear (β = 16.59, *SE* = 1.92, *p* < 0.001) and quadratic terms (β = 3.19, *SE* = 1.59, *p* = 0.045), indicating a positive effect of prior fatigue (linear term) that is particularly strong at extreme levels of fatigue (quadratic term). This might be driven by a potential floor effect in the case of the lower extreme (e.g., an increased likelihood of an increase in fatigue when one is experiencing low fatigue at t − 1) and an especially negative impact of very high fatigue on fatigue at subsequent time points. Lagged productivity showed a significant negative linear association with fatigue (β = −3.28, *SE* = 1.44, *p* = 0.022), suggesting higher prior productivity predicted lower current fatigue; the quadratic term was not significant (*p* = 0.722).

Overall, these results do not support the hypothesis of consistently lower fatigue in the structured break groups (Pomodoro, Flowtime) compared with the self-regulated group based on main effects. Instead, fatigue levels were strongly predicted by prior fatigue and prior productivity. Crucially, a significant interaction revealed that fatigue increased more rapidly for the Pomodoro group compared with the self-regulated group as time elapsed since the last break.

### 5.4. Task Completion

Hypothesis 2 predicted that the Pomodoro and Flowtime conditions would show higher levels of task completion compared with the self-regulated break condition. To test this hypothesis, we employed an Analysis of Covariance (ANCOVA). Task completion (the percent of overall completed tasks) served as the outcome variable. The predictors included group, task amount (the number of planned tasks), total duration spent studying, and the aggregated measures of motivation and productivity. Table 5 presents the ANCOVA table. Additional model diagnostics, including correlation matrix, squared semi-partial correlations, and several diagnostic plots for this model, can be found in Appendix C.

The overall ANCOVA model explained approximately 8.7% of the variance in task completion (Multiple *R*^2^ = 0.09, Adjusted *R*^2^ = 0.02), although the *F*-test (*F*(6, 87) = 1.38, *p* = 0.231) indicated that the model did not reach overall statistical significance.

To assess the unique contribution of each predictor after accounting for others, we examined Type II Sum of Squares from the ANCOVA (see Table 5). This analysis revealed that the primary predictor of interest, group, did not have a statistically significant effect on task completion (*F*(2, 87) = 0.16, *p* = 0.854). Among the predictors, productivity emerged as the only statistically significant predictor of task completion (*F*(1, 87) = 5.30, *p* = 0.024). The estimated coefficient of the productivity term (β = 12.31) suggests that holding all other variables constant, a 1-unit increase in the aggregated productivity measure is associated with an increase of approximately 12.31 percentage points in the overall task completion rate.

Therefore, these findings do not support Hypothesis 2. While break-taking techniques per se did not lead to significant group differences in task completion, higher average productivity appears to be an important factor in predicting the percentage of completed tasks. The overall insignificance of the ANCOVA model and the large residual term indicate that these conclusions should be treated with great care.

### 5.5. Flow State

Hypothesis 3 proposed that the flow state, measured by the Flow State Scale (FSS) total score, would be greater in the self-regulated break and Flowtime conditions than in the Pomodoro condition. To test this, we conducted an ANCOVA with the overall flow score as the outcome. The predictors included group, the number of planned tasks, the total duration spent studying, and the aggregated measures of motivation and productivity. Table 6 presents the ANCOVA table. Additional model diagnostics, including correlation matrix, squared semi-partial correlations, and several diagnostic plots for this model, can be found in Appendix C.

The ANCOVA model demonstrated a Multiple R^2^ of 0.31 (Adjusted R-squared = 0.26). An overall *F*-test indicated that the model was statistically significant (*F*(6, 87) = 6.41, *p* < 0.001).

Examining the unique contribution of each predictor using Type II Sum of Squares (see Table 6) revealed that the primary predictor of interest, group, did not have a statistically significant overall effect on flow scores (*F*(2, 87) = 0.26, *p* = 0.774) after controlling for the covariates. However, both task amount (*F*(1, 87) = 7.40, *p* = 0.008) and productivity (*F*(1, 87) = 8.09, *p* = 0.006) emerged as significant predictors. The respective estimated coefficients suggest that a 1-unit increase in task amount was associated with a 0.12 increase in the flow score, while a 1-unit increase in productivity was linked to a 0.56 increase in flow. In contrast, neither total time spent studying (*p* = 0.297) nor motivation (*p* = 0.433) contributed significantly to explaining the variance in flow scores.

Therefore, these findings do not support Hypothesis 3. While the hypothesized group differences in flow were not found, higher numbers of planned tasks and greater aggregated productivity significantly predicted increased flow levels. The overall model accounted for approximately 31% of the variance in flow, indicating a moderate explanatory power for these factors in relation to the flow experience, but the specific break-taking technique assigned did not significantly influence reported flow.

## 6. Discussion

This study examined the effect on motivation, productivity, fatigue, task completion, and flow of taking self-regulated, Pomodoro, or Flowtime breaks during a self-regulated learning session. Furthermore, we examined how these different break-taking techniques influence the trajectory of subjective study experiences over time.

Hypothesis 1, which stated that the Pomodoro and Flowtime conditions would show higher levels of motivation and productivity and lower levels of fatigue compared with the self-regulated condition, was not supported by our findings. No significant difference in motivation, productivity, or fatigue was found between the break-taking techniques.

However, it was found that the motivation of students who used the Pomodoro and Flowtime techniques declined faster compared with the self-regulated group. This might be explained by the fact that students in these conditions were not allowed to regulate the duration of the break by themselves. This could mean that the breaks are of suboptimal length for some students, which may lead to frustration. Consequently, this may in turn lead to more demotivation ([10]).

Furthermore, it has been found that students who took breaks according to the Pomodoro group showed a faster increase in fatigue levels. This could again be the case because students were not allowed to plan the duration of the breaks in this condition, which may have led to a suboptimal duration of breaks in which students were not able to restore their energy levels sufficiently ([27]; [42]).

Additionally, it was also found that the motivation of students is predicted by prior fatigue, productivity, and motivational levels. Lower fatigue, higher prior productivity, and higher motivation also significantly predicted higher productivity levels. These subjective experiences thus have an influence on each other, which means that it is important for students to restore all of these resources in order to be able to study effectively ([26]; [28]; [31]).

Hypothesis 2, which stated that the Pomodoro and Flowtime conditions would show higher levels of task completion compared with the self-regulated condition, was also not supported by our data. No significant differences in task completion levels between groups were found. However, it seems like productivity is an important predictor of task completion. Students who experienced higher levels of productivity completed a higher percentage of their planned tasks.

The current study findings suggest a minimal impact of different break-taking techniques on subjective study experiences and task completion. Considerable variability among participants within each break condition was observed, likely influenced by several factors. First of all, break effectiveness depends highly on the nature of activities and experiences during the break ([21]; [29]; [35]). Prior research suggests that active breaks, such as engaging in brief physical activities, are found to enhance attention and task performance ([29]; [35]). Furthermore, relaxation and socialization during break time can also enhance break effectiveness ([16]; [21]). In the current study, however, there was no control over the content of the breaks, and no data was gathered on what participants did during the breaks. The variability in the effectiveness of the breaks may therefore be partly explained by a variability in break activities and experiences. The effects of different break activities across these break-taking techniques may be investigated by future research.

In addition, individual differences may also influence the effectiveness of a certain break-taking technique ([15]; [42]). Research shows that people’s level of respite self-efficacy (confidence in recovery during a break) has been linked to break effectiveness in occupational settings. Lower levels of respite self-efficacy are associated with poorer detachment from tasks and reduced break effectiveness ([15]). Moreover, personality traits may influence personal resource depletion and therefore the need for breaks ([14]; [42]). Research suggests that people high in emotional stability experience a reduced need for recovery after work ([14]). Furthermore, extroverts may require fewer and/or shorter breaks than introverts when performing demanding tasks, as they experience lower personal resource depletion ([42]). These findings suggest that the effectiveness of different break-taking techniques may depend on or interact with individual characteristics. This might indicate that different break-taking strategies may work for different people, which may be investigated in future research.

Lastly, the small differences between the break-taking techniques may be explained by the fact that flow was experienced equally in different conditions. The flow state was thus not more disrupted in the Pomodoro condition compared with other break-taking conditions. In the entire sample of this study, flow was not experienced to a great extent. On a scale from 0 to 7, the average level of flow was 4.06 in this sample. A possible explanation for this is that it is important to create a balance between challenge and skill in order to reach a flow state ([43]; [44]). This is because when a task is too easy people will feel bored, but if a task is too difficult people can become stressed, anxious, or feel overwhelmed. In both these cases, a flow state will not be reached ([13]). It could be the case that this balance is not optimal when engaging in study tasks. University assignments are normally more challenging than the current skill level of students ([19]). This is so that students’ skill levels will increase over time. This could explain why, in this sample, the flow state experienced was low in all conditions. The Flowtime technique therefore may not have been found to be more effective than the Pomodoro technique.

### 6.1. Limitations and Future Research

This study comes with several limitations. Due to the authentic learning setting of this study, students could choose when and where to study, which inherently limited experimental control. Students could not be monitored, and therefore, this study setup required the assumption that students would follow the instructions given to them. Another limitation of the authentic learning setting is that all students can choose their own study tasks. This way, not every student worked on a study task of the same difficulty, which might have influenced the dependent variables. Even though we randomized the allocation of students to conditions, the sample size might have been too small to distribute the potential effects of differences in task difficulty across conditions. Future studies might consider keeping the study tasks constant for all students. This will overcome these drawbacks of the authentic study setting, which can make it easier to compare break-taking techniques between participants.

To maintain the authenticity of the study sessions, interruptions were minimized as much as possible. Because of this, single-item measures were used to assess motivation, productivity, and fatigue. Even though these measures may not have been optimal to measure the complexity of these constructs, more extensive measures were avoided to prevent excessive disruption during the students’ study sessions. By using single items, the interruption between study sessions and load for students was also minimized. Nevertheless, students who took more breaks were required to complete the self-evaluations more often, which may have influenced their study and break behavior ([32]). However, if this had significantly affected outcomes, it could be expected that clearer differences between the conditions would have been observed, which was not the case in this study.

Additionally, participants were informed before the study that the study was about break taking. This could have lowered the validity of the study and could have made the students more aware of their break taking ([34]). However, participants were not informed about the different conditions and the goal of the study until after the study was finished, which allowed for minimal interference.

The questionnaire was programmed in such a way that the questionnaire would continue automatically after 2 h of study. However, this function did not work for all participants, probably due to different operating systems on students’ computers and the tasks students performed on their laptops during the study session. Some students therefore needed to manually click a link to the last questions of the survey. This may have led to some limitations, such as students needing to check the time themselves, which could have made them more aware of the time they study.

Furthermore, results regarding feelings of productivity should be treated with care because the proportional hazards assumption might be violated for low levels of productivity. However, given that relatively few participants scored low on productivity, the findings related to productivity levels still hold at higher productivity levels.

Lastly, the current sample introduces some limitations to this study. The final sample of the study, after the removal of outliers, consisted of 94 participants. Because of the shift in analysis approach, this may underpower the analysis, especially for the CLMM analysis. Future research should aim for a bigger sample size to ensure more stable model estimates. Furthermore, the sample of this study is very homogeneous, with approximately 93 percent women, all of whom were psychology bachelor students. Since women are also highly overrepresented in psychology studies, this sample represents the underlying population well ([6]). However, this sample cannot be considered representative of the general student population. We therefore refrain from making generalized statements beyond the specific context of this sample. Future studies should focus on measuring more heterogenous samples so that they can investigate whether the same results are found in other genders and study areas.

Future studies are necessary to investigate the effectiveness of these break-taking techniques further. Future studies can focus on longer study times. By doing this, it can be investigated whether these techniques are efficient for longer sessions. This way it can be investigated whether the development in time on levels of motivation, productivity, and fatigue will continue in a similar manner or not.

Additionally, future studies might focus on mental effort and personal preferences as predictors for the effectiveness of different break-taking techniques to investigate if these are significant contributors to the effectiveness of different techniques.

### 6.2. Conclusions

To conclude, the current findings suggest that for students using Pomodoro and Flowtime techniques, motivation levels decrease faster compared with students who use self-regulated breaks. Similarly, students who used the Pomodoro technique showed a steeper increase in fatigue levels. However, despite these significant differences in slopes of motivation and fatigue over time, these differences did not result in significant differences between the break-taking techniques in the overall means of motivation and fatigue levels. This might indicate that the 2 h study duration was not long enough to find differences among these break techniques. On all other variables measured, no outstanding differences were found in the effectiveness of self-regulated, Pomodoro, and Flowtime breaks. This means that the results of the current study suggest that no one of these break-taking techniques should be more highly recommended over the others. Future research should be performed to investigate the differences between these techniques further by including predictors like personality and mental effort and by examining longer study periods.

## Figures and Tables

**Figure 1 behavsci-15-00861-f001:**
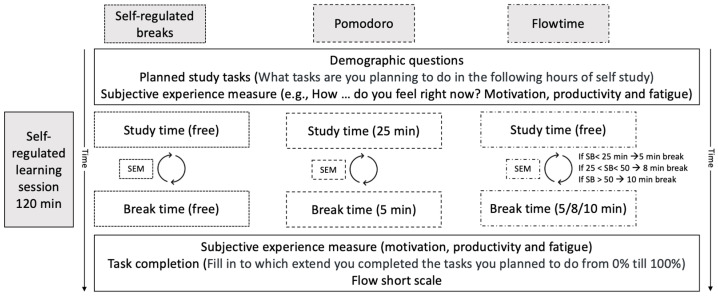
Study procedure for all conditions (self-regulated breaks, Pomodoro, and Flowtime). SEM is the subjective experience measure, SB is study block.

**Figure 2 behavsci-15-00861-f002:**
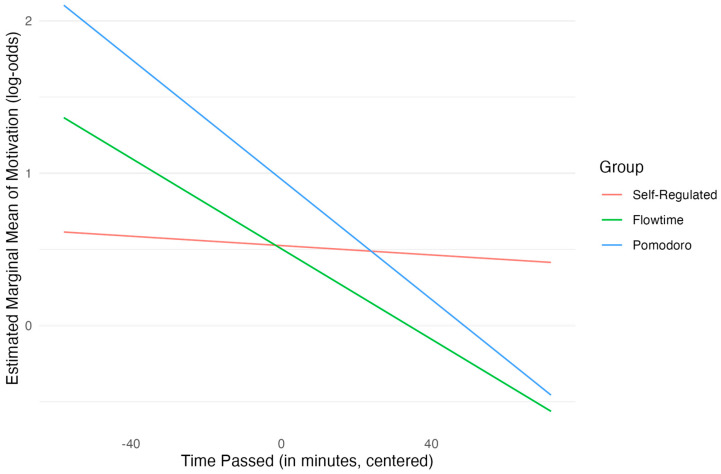
Estimated log-odds of motivation by group over time.

**Figure 3 behavsci-15-00861-f003:**
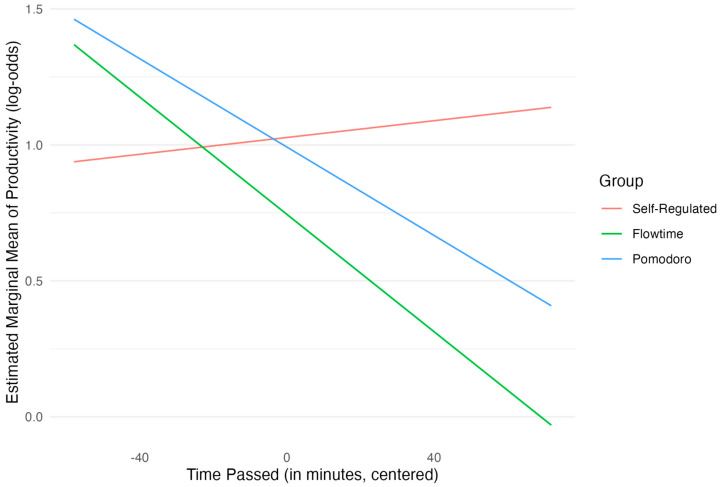
Estimated log-odds of productivity by group over time.

**Figure 4 behavsci-15-00861-f004:**
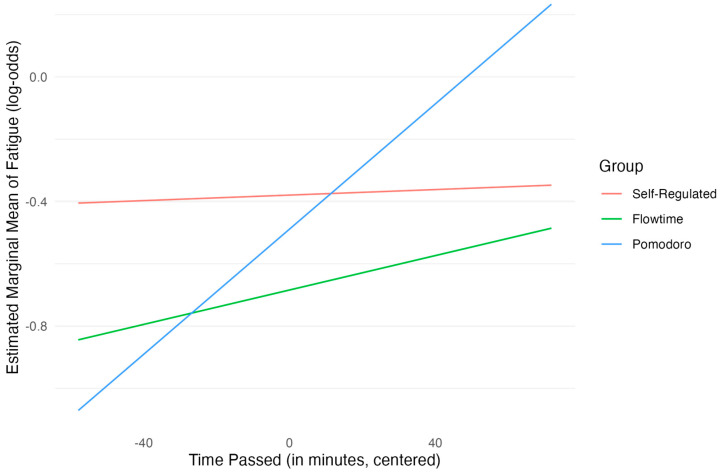
Estimated log-odds of fatigue by group over time.

**Table 1 behavsci-15-00861-t001:** Mean and standard deviation of total study and break time; number of breaks; average motivation, productivity, and fatigue levels; number of planned tasks; total task completion; and flow state in the study for all break-taking conditions separately.

Variable	Self-Regulated Breaks		Pomodoro		Flowtime	
	*M*	*SD*	*M*	*SD*	*M*	*SD*
Total study time	1.71	0.26	1.69	0.03	1.72	0.14
Total break time	0.30	0.26	0.34	0.02	0.30	0.14
Number of breaks	3.04	1.06	4.00	0	2.82	1.65
Average motivation	2.86	0.41	2.73	0.63	2.59	0.57
Average productivity	3.03	0.48	2.86	0.61	2.75	0.61
Average fatigue	2.56	0.50	2.59	0.63	2.64	0.61
Number of planned tasks	3.68	1.93	3.11	1.28	3.73	1.74
Task completion	70.32	20.12	69.41	18.70	67.09	17.50
Flow state	4.22	0.83	4.10	0.74	3.98	0.79

Total study time and total break time in hours; task completion in percentages.

**Table 2 behavsci-15-00861-t002:** Overview of coefficients for motivation levels compared with self-regulated break taking.

Predictor	OR	Log-Odds	Std. Error	*z* Value	*p*-Value
Pomodoro group	1.55	0.44	0.34	1.28	0.200
Flowtime group	0.98	−0.02	0.35	−0.06	0.953
Time (centered)	1.00	−0.00	0.01	−0.33	0.742
Motivation (linear)	55,710.96	10.93	3.76	2.91	0.004 **
Motivation (quadratic)	8.89	2.19	2.62	0.84	0.403
Productivity (linear)	1060.77	6.97	2.95	2.36	0.018 *
Productivity (quadratic)	0.25	−1.37	2.64	−0.52	0.604
Fatigue (linear)	0.01	−4.97	2.73	−1.82	0.069
Fatigue (quadratic)	0.01	−4.43	2.36	−1.87	0.061
Pomodoro × Time	0.98	−0.02	0.01	−2.65	0.008 **
Flowtime × Time	0.99	−0.01	0.01	−2.06	0.039 *

* *p* < 0.05. ** *p* < 0.01.

**Table 3 behavsci-15-00861-t003:** Overview of coefficients for productivity levels compared with self-regulated break taking.

Predictor	OR	Log-Odds	Std. Error	*z* Value	*p*-Value
Pomodoro group	0.97	−0.04	0.30	−0.12	0.904
Flowtime group	0.75	−0.28	0.30	−0.94	0.348
Time (centered)	1.00	0.00	0.01	0.33	0.739
Motivation (linear)	17,281.66	9.76	2.88	3.39	0.001 ***
Motivation (quadratic)	0.28	−1.29	2.48	−0.52	0.602
Productivity (linear)	116,662.27	11.67	3.43	3.40	0.001 ***
Productivity (quadratic)	19.46	2.97	2.55	1.17	0.244
Fatigue (linear)	0.01	−5.26	2.45	−2.15	0.032 *
Fatigue (quadratic)	0.56	−0.57	2.24	−0.26	0.798
Pomodoro × Time	0.99	−0.01	0.01	−1.44	0.150
Flowtime × Time	0.99	−0.01	0.01	−1.96	0.051

* *p* < 0.05. *** *p* < 0.001.

**Table 4 behavsci-15-00861-t004:** Overview of coefficients for fatigue levels compared with self-regulated break taking.

Predictor	Estimate	Std. Error	*z* Value	*p*-Value
Pomodoro group	−0.11	0.19	−0.59	0.558
Flowtime group	−0.31	0.19	−1.57	0.117
Time (centered)	0.00	0.00	0.14	0.885
Last break duration (centered)	−0.04	0.03	−1.37	0.172
Productivity (linear)	−3.28	1.44	−2.28	0.023 *
Productivity (quadratic)	−0.47	1.33	−0.36	0.722
Fatigue (linear)	16.59	1.92	8.66	0.001 ***
Fatigue (quadratic)	3.19	1.59	2.01	0.045 *
Pomodoro × Time	0.01	0.00	2.25	0.025 *
Flowtime × Time	0.00	0.00	0.56	0.577

* *p* < 0.05. *** *p* < 0.001.

**Table 5 behavsci-15-00861-t005:** Overview of ANCOVA model for task completion levels compared with self-regulated break taking.

Predictor	SS	DF	*F*-Value	*p*-Value
Group	105.61	2	0.16	0.854
Task Amount	35.41	1	0.11	0.746
Total Study Time	739.97	1	2.21	0.141
Motivation (Avg.)	669.25	1	2.00	0.161
Productivity (Avg.)	1774.56	1	5.30	0.024 *
Residuals	29,148.23	87	NA	NA

* *p* < 0.05.

**Table 6 behavsci-15-00861-t006:** ANCOVA table for flow state compared with Pomodoro.

Predictor	SS	DF	*F*-Value	*p*-Value
Group	0.23	2	0.26	0.774
Task Amount	3.34	1	7.40	0.008 **
Total Study Time	0.50	1	1.10	0.297
Motivation (Avg.)	0.28	1	0.62	0.433
Productivity (Avg.)	3.65	1	8.09	0.006 **
Residuals	39.23	87	NA	NA

** *p* < 0.01.

## Data Availability

The datasets presented in this article are not readily available because the data was collected as part of a bachelor thesis, and therefore, the ethical approval did not include availability of the data. Requests to access the datasets should be directed to the corresponding author.

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
