# Peer review of "Investigating the Effectiveness of Self-Regulated, Pomodoro, and Flowtime Break-Taking Techniques Among Students"

_behavsci, 2025, doi:10.3390/bs15070861_

Round 1
Reviewer 1 Report
Comments and Suggestions for Authors
The study presents an interesting study on the different pause techniques during self-regulated study. The article is well written and structured.
However, the sample could be larger. The study was also applied to just one two-hour session, which may affect the results obtained.
Additional comments:
The main research question addressed by this study is "What is the effect of the type of break-taking technique (Flowtime, Pomodoro, or Self-regulated breaks) on mental effort, motivation, fatigue, productivity, task completion, and flow state?” The topic of this research is both original and relevant to the field of educational psychology and self-regulated learning.
This study fills a significant gap in the literature by providing a comprehensive, comparative analysis of three break-taking techniques in an authentic self-regulated learning context, with a particular focus on flow state and the temporal dynamics of subjective experiences.
The conclusions are consistent with the evidence presented and adequately address the main research question. The references are relevant to the topics discussed and the tables and figures are well presented.
The authors should re-evaluate the sample size and diverse sample.
Reviewer 2 Report
Comments and Suggestions for Authors
I have carefully read and reviewed the manuscript "Investigating the Effectiveness of Pomodoro, Flowtime, and Self-regulated Break-Taking Techniques among Students," which aims to explore the impact of different break-taking strategies, including Pomodoro, Flowtime, and Self-regulated breaks, on subjective study experiences, task completion, and flow during a two-hour self-regulated learning session among university students. I appreciate the authors for their thoughtful experimental design, particularly the selection of a homogeneous sample of psychology major students and the standardization of study duration, which helps in controlling potential confounding variables and enhances the study's internal validity. However, I have several concerns regarding the statistical models used in the analysis that require further attention. To enhance the clarity and robustness of the findings, I recommend a careful restructuring of the statistical analysis and subsequent revisions of the manuscript. Below, I provide detailed comments and suggestions to assist the authors in refining their approach and presentation, while also welcoming clarification on any points I may have misinterpreted.
Major concern:
- The application of cumulative link mixed effects models (CLMMs) in the manuscript to extend ordinal logistic regression by incorporating random effects is commendable. This methodological choice effectively addresses limitations associated with traditional repeated measures analysis, offering nuanced insights into data characterized by naturally ordered and unevenly spaced categories.
- Circularity and Model Stability: Upon review, I observed a recurring approach where outcome variables such as Fatigue, Motivation, and Productivity are interchangeably used as independent variables across various sections (specifically result sections 5.1, 5.2, and 5.4). This practice raises significant concerns regarding circularity and model stability, as the use of a dependent variable from one CLMM as an independent variable in another model may lead to issues of circularity and instability. This can intertwine the models, potentially reinforcing errors and compromising the reliability of the results.
- Consistency in Data Treatment: I have a concern regarding the treatment of data types across different models, highlighted by the use of the variable 'Productivity'. In sections where 'Productivity' is a dependent variable, it is treated as ordinal and analyzed using a Log-log link function due to its positive skewness. In contrast, in other models where 'Productivity' is used as an independent variable, it appears to be treated as continuous, as suggested by the results presented in the tables, without adjustments for its positively skewed distribution. This inconsistent handling of the same data in various analytical contexts raises significant questions about the methodological consistency of the study. Such variability in data treatment could potentially affect the validity and reliability of the findings. A clarification of the rationale behind these data handling decisions would substantially enhance the robustness and coherence of the analysis.
Given these concerns, would it be possible to focus on one model without rotating the role of these variables across multiple models? Such an approach might provide more clarity and stability to the findings.
- OLS regression models (Section 5.3 & 5.5)
- I appreciate the author's efforts to employ OLS regression with dummy coding and the subsequent reiteration using an alternative dummy code configuration to enable the comparisons posited in their hypothesis (referenced in Sections 5.3 and 5.5). Nevertheless, this strategy appears to simulate the operation of two independent models, which might inadvertently lead to inflation of Type I error, a concern similarly noted with CLMM.
- Might the author consider employing a General Linear Model (GLM) coupled with post-hoc tests across the three groups, incorporating Type I error inflation adjustments such as Bonferroni or Tukey methods? I believe that adopting this methodology could resolve the unusual findings described in Section 5.3—where the overall model did not achieve significance, yet one predictor appeared significantly impactful. This adjustment not only could enhance the robustness and interpretability of the results but also potentially consolidate the analyses for both hypotheses into a singular, more efficient framework.
Minor issues:
- CLMM with Polynomial or Interaction Effects
- The manuscript incorporates models that include higher-order terms, such as quadratic and cubic terms, as well as interaction terms. Within this context, several points warrant attention to ensure the robustness and clarity of the analysis:
- Suggestion for Centering of Terms: The manuscript does not currently employ centering for higher-order and interaction terms, which can lead to multicollinearity, notably seen in the high correlation between linear and quadratic terms. I kindly suggest that the authors consider the potential benefits of centering these terms before incorporating them into the models. This methodological adjustment is crucial for reducing multicollinearity and enhancing the interpretability of the results. Should the authors have theoretical or empirical reasons, supported by citations, for not centering these terms, it would be very informative to include these justifications in the manuscript. Such an inclusion would greatly help readers understand the chosen methodology in the context of existing research.
- Evaluation of Model Complexity and Overfitting: The use of cubic models for the variables Fatigue, Motivation, and Productivity suggests a complex modeling approach. It would be most helpful if the authors could provide empirical justifications or citations from prior studies that support the application of these complex models to these variables. If there are specific reasons or benefits derived from using such models, detailing these would enhance the manuscript’s methodological transparency. Conversely, if simpler models could suffice, considering them might help prevent potential overfitting.
- The manuscript incorporates models that include higher-order terms, such as quadratic and cubic terms, as well as interaction terms. Within this context, several points warrant attention to ensure the robustness and clarity of the analysis:
- Enhancing the Interpretation of Significant Effects: The manuscript briefly describes significant findings related to quadratic and interaction effects in Sections 5.1 and 5.4. I would be particularly interested in a more detailed discussion of how the quadratic terms influence the results and the nature of significant interaction effects, including illustrations of simple slopes or the varying impacts at different time points. Although I recognize that the models used in this study differ from those typically discussed in "Applied Logistic Regression, 3rd edition" by Hosmer, Lemeshow, and Sturdivant, the fundamental concepts applicable to ordinal data in that text could prove beneficial. Exploring this resource might provide valuable insights into effectively presenting and interpreting interaction effects within the framework of your analysis. I recommend considering this text as a potential guide to enhance the depth and clarity of the results presented in your study.
- OLS Regression Analysis in Section 5.3
- The manuscript presents results from an OLS regression analysis in Section 5.3, where the overall model did not reach statistical significance, yet one predictor was found to be significant. This observation, coupled with an effect size ranging from small to medium, raises questions about the underlying statistical relationships and the model's diagnostics. It would be greatly appreciated if the authors could provide a correlation matrix and residual plots to enhance transparency and allow a more thorough assessment of the model's robustness. These could either be included in the appendix or presented in a dedicated table within the manuscript, adhering to historical APA guidelines that recommend disclosing correlations, means, and standard deviations when regression models are utilized. Furthermore, incorporating squared semi-partial correlations in Table 4 would offer valuable insights into the unique contribution of each predictor in explaining the variance of the outcome variable. This addition would not only align with best practices in reporting but also clarify the distinct impact of each predictor within the context of the model. By addressing these points, the authors would significantly enhance the clarity and depth of the analysis presented in Section 5.3.
Miscellaneous:
- Lines 53 (and many others): To correctly cite two articles by Biwer et al. from 2023 in APA style, differentiate them by including the second author's name in each citation: (Biwer, de Bruin, et al., 2023) for the first article, as the second author's name differentiates it from the second citation. (Biwer, Wiradhany, et al., 2023) for the second article, using the second author's name to distinguish it from the first citation.
- Lines110-112: The author may need to review their summary of the findings from Biwer et al., as the statements appear contradictory. The first sentence suggests that students in systematic break conditions reported higher concentration levels and found learning tasks easier compared to those in the self-regulated group. However, the second sentence indicates that students in systematic break conditions experienced lower concentration and motivation compared to those in the self-regulated condition.
- Lines 168-169: I appreciate the authors' transparency in indicating that GPower was used to calculate the initial required sample size. This detail adds value by clarifying the methodological rigor and planning stages of the research. However, I would kindly suggest that the authors consider omitting the detailed explanation of using GPower for computing a priori power. While such calculations are indeed valuable during the proposal stage, it may not be necessary at this point, particularly as there has been a change in the analysis techniques.
- Line 204: I was under the impression that the scale was a 5-point scale, ranging from 1 (not at all) to 5 (very much), rather than a 4-point scale. I might be mistaken. If my understanding is incorrect, could the author please explain why the scale was modified?
- Lines 218-229: Although the author mentions that the measures are provided in Appendix A, it would be beneficial for readers like myself if a brief summary of how the scales function were included in the main text. For instance, the Flow Short Scale (FSS) is a 10-item questionnaire designed to assess the state of flow, which is a psychological condition of profound engagement and enjoyment in an activity. It utilizes a 7-point Likert scale ranging from 1 (not at all) to 7 (very much).
- Line 238: The author stated that "all participants were assigned to one of three conditions," and I assume this assignment was random. If this is the case, I suggest that the author explicitly add "randomly assigned" for clarity.
- Lines 274-281: Once again, I appreciate the authors' transparency in detailing how participant data were excluded from further analysis. However, some of the reasons provided for exclusion, such as age being higher than other participants (38 years old) or not taking any breaks (assuming they were in the self-regulated group), seemed unusual to me. It would be beneficial if the authors could provide a general set of exclusion criteria, offering a clearer rationale for these decisions. While I value the transparency in disclosing the specific reasons for each exclusion, a standardized set of criteria would enhance understanding. For instance, it would be informative to clarify that these participants were removed due to their inability to provide necessary data or because their behavior exceeded the study's predetermined settings, rather than for reasons that might initially seem peripheral to the research objectives. Just a thought.
- Lines 323-324: I noticed that the terms H1a and H1b are used, but I believe this is the first instance they appear in the text. Could you please specify what these terms represent before using them? This clarification would help ensure that the reader fully understands the hypotheses being discussed.
- Lines 320-344: I understand that the author intended to provide examples for the linking function, but it seems that the linking function and random effect structures are thoroughly explained in the subsequent paragraph. Therefore, I recommend that the author consider removing those examples, particularly the sentences within parentheses, to streamline the text. In addition, it would be helpful if the author could explain why specific link functions and random effect structures were applied in the following paragraph. For instance, was the Log-log link used to handle violations of the symmetric shape assumption, perhaps due to a left-skewed distribution indicating that most responses fell into lower categories? This additional explanation could provide clarity on the methodological choices made.
- Line 361 Table 1: In Table 1, it may be beneficial to add the unit of measurement for certain variables in the footnote, such as "Total Study Time," to enhance clarity.
- Line 387: There seems to be a typo regarding the reported value of exp(–0.4794). The calculation should result in approximately 0.61915, but the author reported it as ≈ 0.992.
- Tables 2, 3, 5: I suggest that instead of calculating and discussing the exp(estimated coefficient) in the text, the author could directly present the odds ratios for all variables in the table. This would streamline the presentation and make the data immediately accessible and easier to interpret for the readers.
Reviewer 3 Report
Comments and Suggestions for Authors
Summary comments:
This manuscript reports an experiment that examined the effects of three different methods of studying on a variety of immediate-term academically-useful outcomes. The study uses a relatively small convenience sample of psychology students who were overwhelmingly women (93%). It executes a true randomized experiment and uses sophisticated analysis techniques to arrive at its conclusions, which found very few significant outcome differences between the three studying methods, but that produced more information regarding intermediate effects in the process of studying.
While it has many commendable features, the manuscript evinces several notable limitations. The measures used to assess many of the outcomes are single-item, low-response-option scales that complicate analysis. However, the authors do an admirable job of performing the analysis demanded by that limitation. The authors also note the evident limitations regarding sample size, composition, and data scarcity, and are careful not to overstate or overgeneralize their conclusions. Finally, the authors note the possible confounding effects of letting participants choose their own study material and break activities. There are several other limitations that the authors do not currently note (listed in the line notes below) that should be included in their discussion of the study’s limitations.
Despite these specific impediments, and the relative lack of surprising or unexpected findings forwarded in this manuscript, I believe it can still make a useful contribution to the research corpus regarding this particular phenomena, and recommend it for resubmission with minor revisions.
General comments:
There are many uncontrolled, non-uniform factors involved in this experiment (e.g., difficulty of studying material, different degrees of difficulty/size in self-defined ‘study tasks’, etc.). You are really leaning heavily on randomization to make those effects uniform across groups, and I don’t know that you have quite as many as you need for that purpose (25-36 per group).
Line comments:
11-13: correct capitalization to APA standard here, and throughout the manuscript
16-18: be consistent in spacing throughout (n=25; n= 36) and thoroughly proofread the manuscript for mechanics errors and proper APA style adherence throughout. I leave the rest of this task to the authors.
41-42: limited how? Be specific.
69-74: very repetitive and awkwardly written. Be clear and concise.
87-9: If the break timing and duration is externally and systematically imposed, then is that not by definition not a self-regulated learning setting? This is unclear to me.
91-6: again, repetitive.
91-104: you state that the Pomodoro is a well-known and popular technique. Add citations to support those assertions, preferably with specific numbers so the reader can understand what ‘popular’ means. 5%? 50%?
160-4: if you have research goals beyond those listed in your research questions, make additional research questions.
170-3: We probably don’t need this. Just report the actual achieved power in your results.
174-77: Unnecessary -we assume this.
178-82: Make more general and concise please.
187-8: how many pomodoro sessions were experienced in total for each participant? This influences whether or not they get the 15 minute break at the end of 2 hrs of studying.
191: I don’t follow. Check textual completeness.
202-6: I would like to see stronger measures than single items on a very limited 4-point scale for complex phenomena like motivation. There is a large literature on each of these subjects.
214-6: I need more information here. How did Qualtrics measure this, precisely? Were the students continually interacting with Qualtrics during the sessions?
221: No measure ‘is valid’ -it merely has validity evidence for certain applications and populations.
227-9: Wouldn’t informing the students about the nature of the research cause those with stronger social representation bias (e.g., those who want to look like ‘good students’ or ‘academically competent’) to alter their studying behavior? Wouldn’t this reduce the external validity of the study? If so, this should be listed in the study limitations.
255-61: Those who took more frequent breaks would be forced to fill out more scales than those who took fewer breaks. They would therefore have been forced to do more self-evaluation and metacognitive assessment than others, which very well might have reduced the effectiveness of their breaks. How do you account for this?
282: If your analytic sample was 93% women, then you can’t make any statements about the effects of breaks on non-women. It would be better removing all non-women from the analysis and specify that this is a study of break effects on women psychology students.
297: Usually one also uses BIC and AICc/CAIC to determine relative model fit.
325-7: I thought this might happen. Explain more fully how you simplified your random effects structures please.
328: incomplete text here.
Comments on the Quality of English LanguageThis paper is in need of thorough proofreading with special attention paid to the formatting of the tables, incomplete sentences, and areas of repetitive phrasing.
